# Online Domain Adaptation for Rolling Bearings Fault Diagnosis with Imbalanced Cross-Domain Data

**DOI:** 10.3390/s22124540

**Published:** 2022-06-16

**Authors:** Ko-Chieh Chao, Chuan-Bi Chou, Ching-Hung Lee

**Affiliations:** Department of Electrical and Computer Engineering, National Yang Ming Chiao Tung University, Hsinchu City 300, Taiwan; itria40470@itri.org.tw (K.-C.C.); marco.en09@nycu.edu.tw (C.-B.C.)

**Keywords:** domain adaptation, imbalanced cross-domain data, domain transfer, ANFIS

## Abstract

Traditional machine learning methods rely on the training data and target data having the same feature space and data distribution. The performance may be unacceptable if there is a difference in data distribution between the training and target data, which is called cross-domain learning problem. In recent years, many domain adaptation methods have been proposed to solve this kind of problems and make much progress. However, existing domain adaptation approaches have a common assumption that the number of the data in source domain (labeled data) and target domain (unlabeled data) is matched. In this paper, the scenarios in real manufacturing site are considered, that the target domain data is much less than source domain data at the beginning, but the number of target domain data will increase as time goes by. A novel method is proposed for fault diagnosis of rolling bearing with online imbalanced cross-domain data. Finally, the proposed method which is tested on bearing dataset (CWRU) has achieved prediction accuracy of 95.89% with only 40 target samples. The results have been compared with other traditional methods. The comparisons show that the proposed online domain adaptation fault diagnosis method has achieved significant improvements. In addition, the deep transfer learning model by adaptive- network-based fuzzy inference system (ANFIS) is introduced to interpretation the results.

## 1. Introduction

In the era of the “Industry 4.0” revolution, the stability and reliability of the mechanical equipment is the key to maintaining consistent product quality and whether small faults can be diagnosed in time is a necessary way to ensure the function of the entire mechanical system and avoid total failure. Bearings are one of the most important components of mechanical equipment. The bearing fault may lead to serious safety issues. In recent years, many machine learning technologies have been widely and successfully used in the field of bearing fault diagnosis [1,2,3,4,5]. However, the traditional machine learning methods rely on the training data and testing data are taken from the same domain, such that the feature space and data distribution are the same. Otherwise, the prediction accuracy of these fault diagnosis models may be severely reduced. In fact, it is very hard to collect training data that matches the feature space and data distribution of the testing data in real world applications.

In rolling bearing fault diagnosis cases, the data used for training the classification model may be collected and labeled from the motors without any load, but in practical application is to detect the fault of the rolling bearing under various motor load conditions (not zero). In spite of the fact that categories of faulty are the same at different work conditions, the target data features distribution become different with the input data. Accordingly, if the classification model that built by the training samples is applied directly to the target samples, its performance will significantly decline. Moreover, it is costly or even impossible to recollect all kinds of faulty data and label them at different work conditions for reconstructing or fine-tuning the fault diagnosis model. Therefore, there is a need to create reliable and accurate methods to train with data from different domains but related [6,7].

Recently, many domain adaptation methods focus on transferring information from source domain (labeled data) to target domain (unlabeled data) by mapping the data into shared feature space and minimizing the distance between the feature distributions of both domains. Such as Bregman divergence [8], Kullback–Leibler divergence [9], and Maximum mean discrepancy (MMD) [10] have been widely adopted as a discrepancy metric between source and target domains. For rolling bearing fault diagnosis under different working conditions, MMD is utilized to reduce the domain discrepancy between distributions via mapping the data into a Reproducing Kernel Hilbert Space (RKHS) [11]. Although domain adaptation methods have many advantages, it still suffers from limited target domain data. When the target data is much less than source domain data, there may not be enough information to calculate the domain discrepancy, which would cause the performances of these approaches to drop significantly. Nevertheless, insufficient data in the target domain is very common in real-world applications, so a new domain adaptation method especially for imbalanced cross-domain data is very necessary to provide [12].

In this study, a novel bearing fault diagnosis framework is proposed by considering the characteristics of fault diagnosis problems and the situation of imbalanced cross- domain data, which is based on a combination of the convolutional neural network (CNN), the short time Fourier transform (STFT) and Maximum mean discrepancy (MMD). Our proposed method consists of three modules: data preprocessing, condition recognition and domain adaptation. The data preprocessing module used STFT to convert vibration signals into corresponding time-frequency maps (2D). The condition recognition module used CNN to automatically learn features and accurately recognize the conditions of bearings. Last but not least, the domain adaptation module is used as MMD as loss function to make the features extracted by CNN become more representative in target domain. The Case Western Reserve University bearing dataset [13] was then used to verify our proposed method. The results show that STFT is an effective way to cope with limited target domain data and is beneficial to the follow-up CNN training. Since the time- frequency maps having the information of both time domain and frequency domain at the same time, which facilitates the CNN to learn features from limited data. Also, the results indicate that the proposed method improves the prediction accuracy about 5.23% compared with traditional methods in the situation of only 40 target samples. Finally, the deep transfer learning model by replacing the fully-connected layers as adaptive neuro- fuzzy inference system (ANFIS) to provide the interpretation of classification.

The main contributions of this literature are summarized as follows:(1)A novel bearing fault diagnosis framework is proposed. The characteristics of fault diagnosis problems and the situation of imbalanced cross-domain data are both considered.(2)Replace the fully connected layers with the so-called adaptive neuro-fuzzy inference system (ANFIS) by transfer learning. In order to improve the lack of transparency and interpretability in ML model.(3)As a result, our proposed method achieves a significant improvement by comparing with other traditional methods in the situation of few target samples.

In the rest of this paper, preliminaries, including short-time Fourier transform, convolutional neural network (CNN), maximum mean discrepancy (MMD) and ANFIS, are introduced in Section 2. Section 3 introduces the proposed method for online domain adaptation with imbalanced data. Experiments and analysis using the CWRU dataset are presented in Section 4. Finally, the conclusion is given in Section 5.

## 2. Preliminaries

*A.* 
*Short-Time Fourier Transform*


Short-time Fourier transform (STFT) is a signal analysis method that contains time-domain and frequency-domain information, which is specially suitable for analyzing time-varying, non-stationary signals. The visual representation output by STFT is called spectrogram, which adds time-domain information to the fast Fourier transform (FFT). Most importantly, STFT could transform raw vibration signals (1D) into pictures (2D), which are more suitable for the following CNN to process. The basic formula of STFT is defined as follows:(1)$$STFT\left\{x\left(t\right)\right\}\left(\tau ,\omega \right)=\int_{-\infty }^{\infty }x\left(t\right)\omega \left(t-\tau \right)e^{-i\omega t}dt$$
where $x\left(t\right)$ is the signal to be transformed, ω denotes the frequency, and $\omega \left(\tau \right)$ is the window function. The frequency resolution and time resolution of the spectrogram could be determined by changing τ. For instance, the shorter length of window function provides higher time resolution and lower frequency resolution. In this study, STFT were used to convert vibration signals into corresponding time-frequency maps (2D).

*B.* 
*Convolutional neural network and Batch Normalization*


Convolutional Neural Network (CNN) proposed in 1998, it is always utilized for classification and prediction in image processing and other researching fields [14]. Figure 1 shows the illustrated network structure of a classical CNN, generally speaking, CNN architecture is mainly composed of three parts: (1) convolutional layer; (2) pooling layer; and (3) fully-connected layer. In this study, CNN were designed to automatically learn features and accurately recognize the conditions of bearings. In addition, batch normalization layers were added to the neural network in order to improve the performance of CNN [9].

The convolutional layer contains many filters, and each filter contains different inside values that can be convolved with the input data to detect different kinds of features. As stated above, the main function of the convolutional layer is to find out what the important features of the input are data by learning the weights of every filter.

The pooling layers often follow convolution layers. It can be seen as a down-sampling method, which reduces the feature map dimensions of the previous layers. As stated above, the main function of the pooling layer is to reduce the computing costs of the neural network.

The batch normalization (BN) layer usually inserts between the convolutional layer and the pooling layer. BN is a method used to make neural networks converge faster and more consistently when training by re-centering and re-scaling the inputs from different layers. Ideally, the resulting normalized activation has zero mean and unit variance. In this study, we adopt BN right after each convolution and before activation, following [15]. The input of the batch normalization layer is $X\in R^{m\times k}$, where k and m denote the feature dimension and the batch size, and the BN layer transforms each feature $i\in \left\{1\ldots k\right\}$ into:(2)$${\hat{x}}_{i}=\frac{x_{i}-E [X_{j}]}{\sqrt{Var [X_{j}]}},\;\;y_{j}=\gamma_{j}{\hat{x}}_{i}+\beta_{j}$$
where $x_{i}$ is an input feature and $y_{j}$ is the corresponding output,$\;X_{j}$ denotes the jth layer of network, and $\gamma_{j}$ and $\beta_{j}$ control the scale and shift of the input to retain data diversity, which is determined by training.

The fully-connected layers are used as a classifier at the end of CNN architecture to classify the extracted features. The fully-connected layers consist of multiple hidden layers, which is equivalent to the neural network. In general cases, the features extracted by convolution layer must be sent to the fully connected layer to complete the final operation of the model.

*C.* 
*Maximum Mean Discrepancy*


Maximum mean discrepancy (MMD) is a criterion used to estimate the difference between two probability distributions. MMD is defined as the squared distance of the mean embedded features in the reproducing kernel Hilbert space (RKHS). MMD only becomes zero if (and only if) the two distributions are the same. In this paper, MMD is used to calculate the domain discrepancy between the source domain $\left(x^{s}\right)$ and the target domain $\left(x^{t}\right)$. Suppose the probability distribution of the source domain data is *P* and that of the target domain data is *Q*, MMD could be defined as:(3)$$\left(f,P,Q\right)={sup}_{f\in F}\left(E_{x_{s}∼P}\left[f(x^{s})\right]-E_{x_{t}∼Q}\left[f(x^{t})\right]\right)$$
where f: x →H is a projection function. As we choose *f*, which is the unit ball in a universal RKHS, Equation (3) can be rewritten as:(4)$$D\left(X^{S},X^{T}\right)=∥\frac{1}{n_{S}}\overset{n_{S}}{\underset{i=1}{\sum }}\phi \left(x_{i}^{\left(S\right)}\right)-\frac{1}{n_{T}}\overset{n_{T}}{\underset{j=1}{\sum }}\phi \left(x_{j}^{\left(T\right)}\right){∥}_{H_{k}}$$
where $H_{k}$ denotes the RKHS with a characteristic kernel k, which is related to the feature map ϕ. MMD is calculated by the kernel method for practical application, which originally came from SVM. The kernel function can be defined as $k\left(x^{s},x^{t}\right)=\langle \phi \left(x^{s}\right),\phi \left(x^{t}\right)\rangle$.

Kernel choice is also very important as it will affect the performance of MMD. According to [16], multi-kernel MMD which use different kernels for ensuring is one of the best kernel choices. A multi-kernel MMD function consisted of $N_{k}$ radial basis function kernels are shown below:(5)$$k\left(x^{S},x^{T}\right)=\overset{N_{k}}{\underset{j=1}{\sum }}\;k_{\sigma j}\left(x^{S},x^{T}\right)$$
where $k_{\sigma i}$ is the Gaussian kernel and σi its corresponding bandwidth. In this paper, the MMD is adopted for domain adaptation.

*D.* 
*Adaptive Neuro-Fuzzy Inference System*


Adaptive neuro-fuzzy inference system (ANFIS) [17] is a combination of artificial neural networks (ANNs) and fuzzy inference systems (FIS). ANNs are models that are usually referred to as black boxes, because they are too complex or deep for a human to understand how the model achieved its goal. For lots of real-world problems, ANNs have a big advantage by not requiring physical pre-information before training a model, but their utility has been critically limited due to the interpretation that the “black box” model is difficult. In contrast, the FIS model is like a white box, it provides the fuzzy logic rules of human thinking for decision making with imprecise and non-numerical information, i.e., the model designers can figure out how it works. All in all, ANFIS applies the ANNs technique to compute the parameters of a fuzzy model automatically and the outputs map out into the fuzzy model can be explainable. In this study, ANFIS is adopted to replace the fully-connected layers (last two layers) of the network in order to provide reliable prediction and understand the mechanism underlying the algorithms, which can make artificial intelligent methods more transparent.

The architecture of ANFIS is shown in Figure 2. Five layers are used to construct this model. For simplicity, we assume the fuzzy inference system under consideration has two inputs $x_{1}$ and $x_{2}$ and one output f. Suppose that the rule base contains two fuzzy if-then rules of Takagi and Sugeno-type [18].
$$Rule\;1:\;If\;x_{1}\;is\;A_{1}\;and\;x_{2}\;is\;B_{1},\;then\;f_{1}=\alpha_{1}x_{1}+\beta_{1}x_{2}+\gamma_{1}$$
$$Rule\;2:\;If\;x_{1}\;is\;A_{2}\;and\;x_{2}\;is\;B_{2},\;then\;f_{2}=\alpha_{2}x_{1}+\beta_{2}x_{2}+\gamma_{2}\;$$

***Layer 1:*** The first layer is used to convert the inputs into a fuzzy set by membership functions (MFs).
(6)$$O_{1i}=\mu_{A_{i}}\left(x_{1}\right),\;\;for\;i=1,\;2\;O_{2i}=\mu_{B_{i}}\left(x_{2}\right),\;\;for\;i=1,\;2$$
where $x_{1}$ and $x_{2}$ are the input nodes i, A and B are the linguistic labels (small, large, etc.) associated with this node function. Usually, we choose $\mu_{A_{i}}\left(x_{1}\right)$ and $\mu_{B_{i}}\left(x_{2}\right)$ to be Gaussian-shaped functions, where MFs have maximum and minimum values equal to 1 and 0, respectively. In other words, $O_{1i}$ is the degree to which the given $x_{1}$ satisfies the quantifier $A_{i}$ and $O_{2i}$ is the degree to which the given $x_{2}$ satisfies the quantifier $B_{i}$.

***Layer 2:*** The second layer represents the multiplication of the incoming signals and sends the product out.
(7)$$W_{i}=\mu_{A_{i}}\left(x_{1}\right)\times \mu_{B_{i}}\left(x_{2}\right)=O_{1i}\times O_{2i},\;\;for\;i=1,\;2$$
where the output signal $W_{i}$ represents the firing strength of the rule.

***Layer 3:*** The third layer is used to normalize the firing strength by computing the ratio of the ith node firing strength to the sum of all the rules’ firing strengths.
(8)$$\underline{W_{i}}=\frac{W_{i}}{W_{1}+W_{2}},\;\;for\;i=1,\;2$$
where the $\underline{W_{i}}$ is the normalized firing strength.

***Layer 4:*** The fourth layer represents that each node function multiplied by its weight value.
(9)$$O_{4i}=\underline{W_{i}}\times f_{i},\;\;for\;i=1,\;2$$
where $f_{1}$ and $f_{2}$ are the fuzzy if–then rules as mentioned above.

***Layer 5:*** The last layer is used to compute the overall output as the summation of all incoming signals.
(10)$$O_{5i}=\underset{i}{\sum }\underline{W_{i}}\times f_{i}=\frac{\sum_{i}W_{i}f_{i}}{W_{i}}=Overall\;output$$

## 3. On-Line Domain Adaptation

### 3.1. Model Architecture

According to the results of [19], STFT has great potential to preprocess vibration signals and is beneficial to the follow-up CNN training. In this study, the architecture of the proposed CNN model is shown in Figure 3. At first, the vibrational signals (raw data) are transformed into STFT time-frequency spectra, the raw vibrational signal is transferred into image. Then, a CNN model with seven layers including one input layer, two convolution layers, two pooling layers, one fully connected layer, and one output layer, is trained to classify the bearing conditions. The detailed network structure of the CNN is introduced in Table 1. Finally, a domain adaptation model is trained to reduce the distribution distance between two domains (target and source). Please note that a large imbalanced data ratio between source and target sets is considered for the manufacturing site. In addition, the cross-entropy (CE) loss is utilized for the training of source data and the MMD loss is adopted to minimize the distribution difference.

### 3.2. Optimization Objective

CNN architecture is used to extract the discriminative features of different classes. Therefore, the cross-entropy loss (CE) function $L_{c}$ is considered as a term of loss function to minimize the classification error on the source domain.

Additionally, for narrowing the distribution distance between two domains, MMD between source and target samples, is also considered as a term of loss function. The multi-kernel MMD loss could be rewritten as follows from Equation (5):(11)$$L_{m}=\Sigma_{k\in K}MMD_{k}\left(f^{S},f^{T}\right)$$
where $f^{S}$ and $f^{T}$ are the source and target domain features’ representations in the last fully connected layer of the network. In this study, five kernels $K\in \left\{1,2,4,8,16\right\}$ were chosen and set as the same weight because of its high performance [12].

Combining cross-entropy loss with maximum mean discrepancy loss as a total objective function, the network could learn not only to capture the domain invariant features between two domains but also to extract the discriminative features of different classes. That is, the model would classify well in both of the two domains as the loss function converges. Hence, the overall objective function is represented using Equation (6) as:(12)$$L_{total\;}=L_{c}+\beta L_{m}$$
where β is penalty coefficient, we set its value to 1 in the whole study for simplicity.

### 3.3. General Procedure of the Proposed Method

In this study, a novel method is proposed for fault diagnosis of rolling bearing with online imbalanced cross-domain data. The general procedure for proposed method is shown in Figure 3 and the basic steps are described as follows.

***Step 1*:** The vibration data of rolling bearing is measured by acceleration sensors in real application, herein we use CWRU dataset to instead this part for testing. The state of bearings is classified as normal, ball fault, inner raceway fault, and outer raceway fault.

***Step 2*:** In the preprocessing stage, STFT were used to convert vibration signals into corresponding time-frequency spectra (2D). The results of STFT is shown in Figure 4.

***Step 3*:** In the training stage, the CNN model is constructed for classify the bearing conditions from the time-frequency spectra. The initial learning rate is 0.0001, the optimizer is Adam, and loss function is using eqs. (12). Calculate the loss of the CNN model and update the parameters until the stopping criterion is reached

***Step 4*:** Target domain testing samples are used to authenticate our proposed method.

## 4. Experiments and Results

In this section, the proposed method is conducted on a rolling bearing fault dataset, and the detailed information of the hyperparameters can be found in Table 2. The models are written in Python by using the PyTorch repository on the GPU (NVIDIA GeForce GTX 1660).

### 4.1. Dataset Description

The rolling bearing dataset used in this study was provided by the Bearing Data Center of Case Western Reserve University (CWRU). The dataset has been widely used in related research works [20,21,22,23,24]. The CWRU bearing vibration data were collected from the drive end of the motor through an accelerometer at 12,000 samples/second. There was one healthy condition and three fault types (ball fault, inner raceway fault, and outer raceway fault) with three different damage sizes (0.007, 0.014, and 0.021 inches), which provide the experiments with 10 categories of classification tasks. The experiment was also repeated under different motor loads, including 0, 1, 2, and 3 horsepower (hp). At each load, motors have different rotational speeds, which could be considered as different domains.

Raw data were split into specific shapes: 2400 samples of 500 sample length without overlapping. The preprocessing procedures on the source domain data (labeled) and the target domain data (unlabeled) are the same. In the first experiment, the vibration signal with 0 hp load is chosen as the source domain, and 3 hp load is chosen as the target domain. The number of target domain data for the domain adaptation training is gradually increasing (from 0 to 2000), but that of the testing data remains unchanged (400). As for the source domain data, it remains 2000 for the entire experiment. Each condition was trained 10 times and the average and standard deviation were calculated, which could be used to represent the accuracy and stability respectively.

### 4.2. Results and Discussion

The classification average accuracy and standard deviation results of the proposed method (STFT + CNN) are summarized in Table 3 and the results of the traditional method (FFT + NN) are shown in Table 4. Through the results, we obtain observations as follow:(1)When the number of target domain data reaches 26, the accuracy of the proposed method will be over 90%, whereas a traditional method needs 40 target data to achieve a prediction accuracy of 90%. When the number of target domain data reaches 150, the accuracy will be over 99% for our proposed method, while a traditional method needs 1000 data to achieve over 99% accuracy.(2)When the target domain data starts out very low (<40), the testing accuracy will increase rapidly as the target data increases. If the target domain data is over 50, the standard deviation of the testing accuracy starts to decrease as the target data increases.(3)The comparison results of the accuracy and standard deviation (STD) of ten independent trials are shown in Figure 5: solid lines denote the results of FFT + NN; and dashed-lines denotes the results of STFT + CNN. Obviously, it can be observed that the proposed method outperforms (higher accuracy and smaller STD) the traditional method for imbalanced cross-domain data.(4)Note that the imbalanced ratio with a value of infinity means that there is no domain adaptation process. This means that the model is trained by source data and then obtains the inference results using target inputs directly. We can observe that the accuracies of STFT + CNN and FFT + NN are both lower (74.54% and 67.84%) than results with domain adaptation. This illustrates the advantage of domain adaptation. In addition, the performance of STFT + CNN is better than the result of FFT + NN, which demonstrate the improved performance of the proposed approach.(5)Figure 6 shows that the proposed method (STFT + CNN) has better performance than traditional cross-domain fault diagnosis methods (FFT + NN) when there is a lack of target domain data (from 0 to 2000). Although the traditional method is good enough (99%) to use in the condition that source domain and target domain data are both sufficient, its accuracy will drop rapidly when two domains data are imbalanced. For example, when we have 40 target samples, the proposed method could reach an accuracy of about 95%, but the traditional fault diagnosis method only has an accuracy of roughly 90%.

To evaluate the performances of the proposed method, another cross-domain fault diagnosis method was adopted for comparison, deep adversarial domain adaptation (DADA) [25], which was designed specifically for a small amount of data. The DADA model uses a domain discriminator to replace the MMD loss to bridge the gap between two domains.

There are experiments on different cross-domain tasks in Table 5. These results once again verified the benefits of the proposed method. The DADA model shows its stability, its accuracy does not drop dramatically as target data decreases. But considering the accuracy when data becomes sufficient, the performance of the proposed method is better than the DADA method. It can even better adapt to the lack of data than the DADA method, which was designed especially for this condition. Moreover, by comparing the results of FFT+NN and FFT+CNN, it can be found that CNN is more adaptable to a small amount of data than NN. Furthermore, from the comparing the results of FFT+CNN (1D) and STFT+CNN (2D), CNN is more suitable for extracting 2D features than 1D features when encountering the lack of target domain data.

### 4.3. Transfer Learning Model by ANFIS

As the mention in literature [26], the trust AI systems should be introduced that are results. FIS based on human expert experience to make decisions with imprecise and non-numerical information, i.e., the If-Then fuzzy rules are designed by human experiences to predict the corresponding output. In literature [17], ANFIS combines the advantages of FIS (human thinking) and ANN (learning ability) which is built by dataset. It has layer structure similar to neural networks but with the operations of fuzzy inference system, e.g., rules layer, defuzzification layer. Herein, ANFIS is utilized to combine with the proposed model (shown in Figure 7). By replacing the last two layers of the network with ANFIS and fixed the parameters of the remaining CNN model which makes the AI model becomes more explainable. As shown in Figure 2, inputs of ANFIS is the flatten variables and each input having two fuzzy membership functions, the first-order Sugeno-type ANFIS is created. We could know why the system outputs this value from the given input by the membership function and fuzzy rule we set. However, some strengths are not always advantageous, the performance of ANFIS is a little worse than artificial neural networks. In this study, combining the ANFIS to the proposed model at the condition of enough target domain data, the corresponding performce in accuracy is about 94%. By the way, the corresponding computational effort is reduced due to the paramaters of CNN are fixed.

## 5. Conclusions

This paper proposed a novel method to solve the online domain adaptation rolling bearings fault diagnosis problem with imbalanced cross-domain data, which is consistent with actual applications. The superiority of proposed method was demonstrated by comparisons with the other peer methods. The proposed method has great potential for handling with imbalanced cross-domain data. Since the time-frequency spectra having the information of both time domain and frequency domain at the same time, which facilitates the following deep neural network to learn features from limited data. The main drawback of this paper was that it assumed that sufficient and well-labeled source domain data are available for training. In real world application, it is costly or even impossible to obtain all kinds of faulty data and label them at precise conditions. Finally, the proposed approach was modified by transfer learning using ANFIS, the corresponding results show well performance in accuracy, in addition, there is a trade-off between model interpretability and model accuracy.

## Figures and Tables

**Figure 1 sensors-22-04540-f001:**
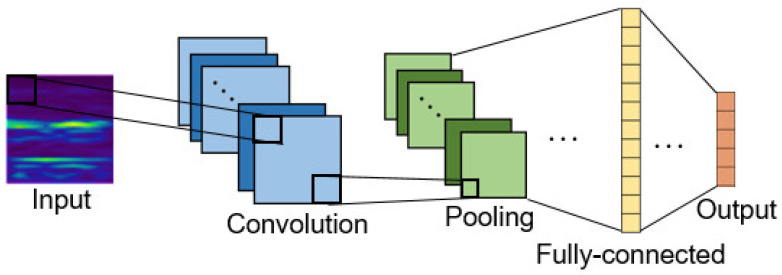
CNN structure [13].

**Figure 2 sensors-22-04540-f002:**
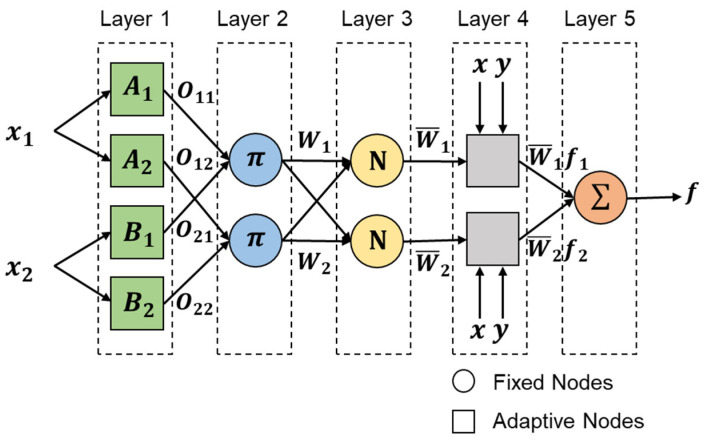
ANFIS structure [17].

**Figure 3 sensors-22-04540-f003:**
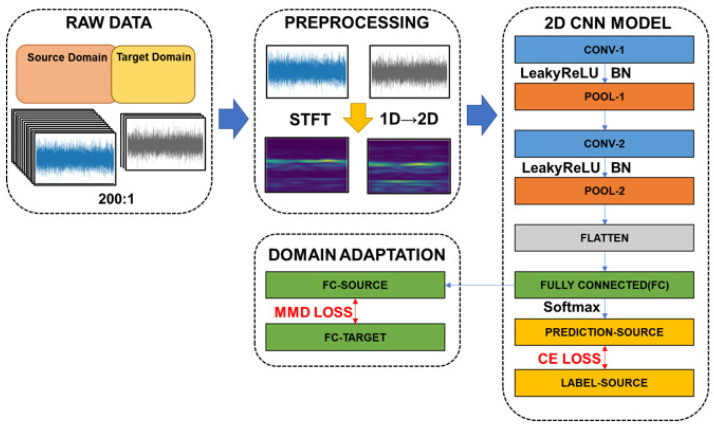
Illustration of the proposed cross-domain fault diagnosis model structure.

**Figure 4 sensors-22-04540-f004:**
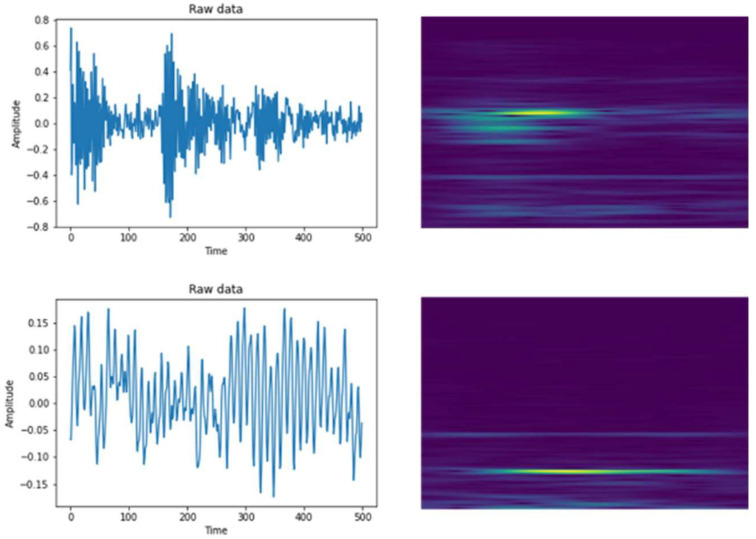
Results of STFT.

**Figure 5 sensors-22-04540-f005:**
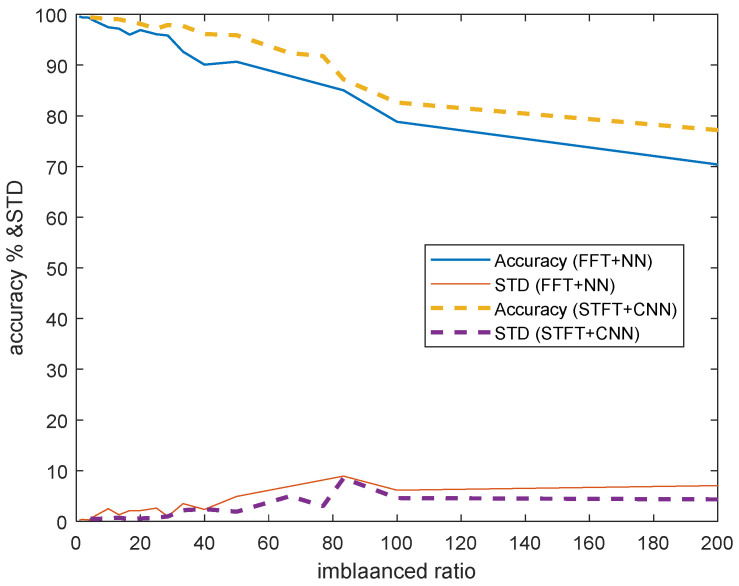
Comparison results of the proposed method (STFT + CNN) with FFT + NN.

**Figure 6 sensors-22-04540-f006:**
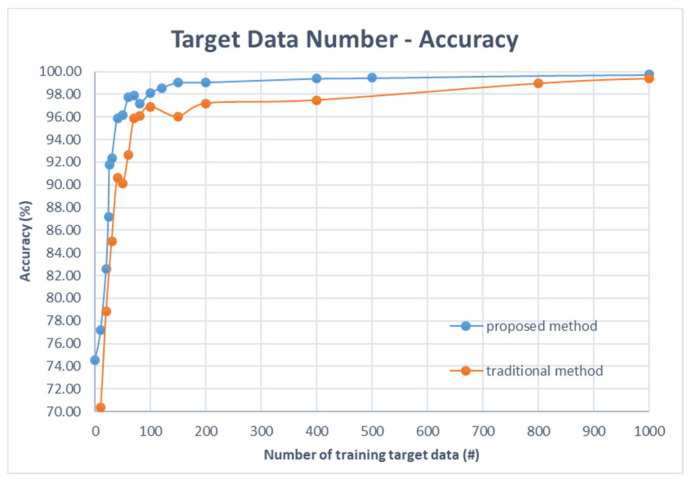
Accuracy of the two models with limited target domain data: blue line is the proposed method (STFT + 2D CNN + MMD loss) and orange line is the traditional method (FFT + NN + MMD loss).

**Figure 7 sensors-22-04540-f007:**
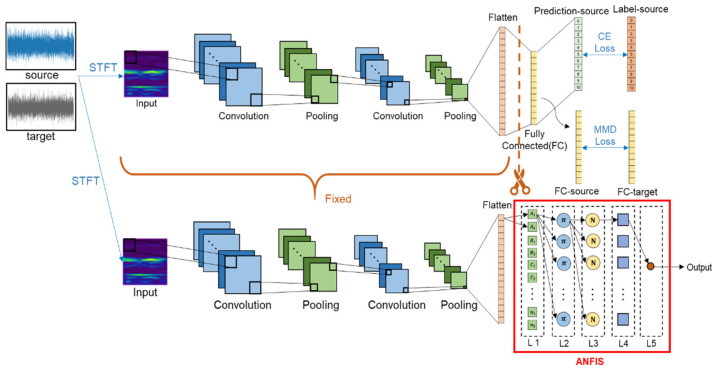
The structure of the proposed model combines with ANFIS.

**Table 1 sensors-22-04540-t001:** Architecture of the proposed method.

Layer Type	Parameters
2D Convolutional Layer (1st)	Filter Number = 6 Filter Size = (9, 9)
Batch Normalization layer (1st)	Activation Function = Leaky ReLU
Max Pooling Layer (1st)	Filter Size = (2, 2)
2D Convolutional Layer (2nd)	Filter Number = 12 Filter Size = (3, 3)
Batch Normalization layer (2nd)	Activation Function = Leaky ReLU
Max Pooling Layer (2nd)	Filter Size = (2, 2)
Flatten	-
Fully Connected Layer	100 neurons
Softmax Output Layer	10 neurons

**Table 2 sensors-22-04540-t002:** Hyperparameters of the proposed method.

Hyperparameters	Value
Epochs	40
Learning rate	0.0001
Optimizer	Adam
Batch size	64
Sample length	500
Source domain training sample	2000
Target domain training sample	0~2000
Target domain testing sample	400

**Table 3 sensors-22-04540-t003:** Results of the proposed method with imbalanced cross-domain data.

Number of Source Domain Data	Number of Target Domain Data	Imbalanced Ratio	Average Accuracy (%)	Standard Deviation (10 Trials)
2000	0 (without DA)	∞ (without DA)	74.54	4.590
2000	10	200	77.17	4.326
2000	20	100	82.61	4.595
2000	24	83.33	87.19	8.518
2000	26	76.92	91.79	2.994
2000	30	66.67	92.36	4.976
2000	40	50	95.89	1.895
2000	50	40	96.13	2.417
2000	60	33.33	97.71	2.194
2000	70	28.57	97.92	0.982
2000	80	25	97.19	0.718
2000	100	20	98.13	0.570
2000	120	16.67	98.56	0.464
2000	150	13.33	99.03	0.714
2000	200	10	99.06	0.561
2000	400	5	99.39	0.480
2000	500	4	99.44	0.644
2000	1000	2	99.71	0.325
2000	2000	1	99.78	0.245

**Table 4 sensors-22-04540-t004:** Results of traditional method with imbalanced cross-domain data.

Number of Source Domain Data	Number of Target Domain Data	Imbalanced Ratio	Average Accuracy (%)	Standard Deviation (10 Times)
2000	0 (without DA)	∞ (without DA)	67.84	3.760
2000	10	200	70.38	7.043
2000	20	100	78.81	6.139
2000	30	83.33	85.03	8.919
2000	40	50	90.66	4.913
2000	50	40	90.09	2.353
2000	60	33.33	92.63	3.493
2000	70	28.57	95.84	0.970
2000	80	25	96.09	2.611
2000	100	20	96.91	2.124
2000	150	16.67	96.00	2.113
2000	200	13.33	97.19	1.304
2000	400	10	97.47	2.496
2000	800	5	98.93	0.768
2000	1000	4	99.36	0.274
2000	1200	2	99.39	0.375
2000	2000	1	99.58	0.274

**Table 5 sensors-22-04540-t005:** Classification accuracy for six different quantity of target domain data.

	Source: 2 hp Target: 0 hp					
Methods	The number of target domain training data					
	20	50	100	500	1000	2000
DADA	87.42%	88.08%	88.17%	88.58%	88.92%	91.25%
FFT + NN	85.92%	93.42%	96.08%	99.17%	99.75%	99.92%
FFT + CNN	93.81%	96.67%	98.83%	99.33%	99.67%	99.92%
STFT + CNN (Proposed)	92.50%	98.25%	98.92%	99.42%	99.50%	99.92%
	Source: 0 hp Target: 2 hp					
Methods	The number of target domain training data					
	20	50	100	500	1000	2000
DADA	84.83%	88.58%	91.17%	91.42%	92.50%	93.00%
FFT + NN	82.17%	92.00%	96.50%	99.75%	99.67%	99.83%
FFT + CNN	90.58%	97.33%	98.42%	99.73%	99.92%	99.92%
STFT + CNN (Proposed)	95.75%	97.42%	99.17%	99.83%	99.83%	100.00%
	Source: 1 hp Target: 3 hp					
Methods	The number of target domain training data					
	20	50	100	500	1000	2000
DADA	95.75%	95.92%	96.67%	97.33%	97.42%	97.75%
FFT + NN	88.75%	93.58%	97.17%	99.17%	99.50%	99.58%
FFT + CNN	94.33%	97.67%	99.00%	99.67%	99.75%	99.92%
STFT + CNN (Proposed)	94.83%	98.50%	98.87%	99.42%	99.83%	99.83%
	Source: 3 hp Target: 1 hp					
Methods	The number of target domain training data					
	20	50	100	500	1000	2000
DADA	81.00%	84.33%	85.50%	86.17%	86.25%	86.83%
FFT + NN	80.67%	89.00%	88.92%	97.33%	99.50%	99.67%
FFT + CNN	87.67%	96.25%	97.00%	99.08%	99.58%	99.67%
STFT + CNN (Proposed)	89.92%	97.00%	97.17%	98.92%	99.58%	99.75%

## Data Availability

Not applicable.

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
