# Peer review of "Online Domain Adaptation for Rolling Bearings Fault Diagnosis with Imbalanced Cross-Domain Data"

_sensors, 2022, doi:10.3390/s22124540_

Round 1

Reviewer 1 Report

The paper under review considers the issue of "Online Domain Adaptation for Rolling Bearings Fault Diagnosis with Imbalanced Cross-Domain Data"

In the reviewer’s opinion, in general, the paper is quite interesting. However, there are several important aspects that require authors comments or possibly improvements:

1) Why did the authors use STFT instead of the Kurtogram?

2) What were the STFT results for the sample data like? Nothing is known about the input data.

3) Bibliography is biased. It mostly contains quotes from Chinese people.

Author Response

We would like to thank you for your insightful comments and valuable suggestions. This paper has been carefully revised according to their constructive comments/suggestions. Two subsections are added in the revised version according reviewers commends,

Section 3.3 introduces the general procedure of the proposed method (pages 7-8);

Section 5 discuss the disadvantages and advantages of the proposed method (pages 13);

In addition, the manuscript has been corrected by Editing office, certification is attached as follows. The details are as follows.

Reviewer 2 Report

In this work is proposed a novel bearing fault diagnosis framework that deals with the characteristics of fault diagnosis problems and the situation of imbalanced cross-domain data. The method is validated through the CWRU experimental data.

The proposal is interesting and the results are convincing, however some issues must be addressed.

1. Please avoid writing in a personal way such as "... we applied the classification model which id built..."

2. I suggest to provide the description of Section 2 according to the designed application, it could improve its understanding

3. If Section 3 aims to depict the proposed method it needs more information. Please include more details.

4. What are the disadvantages and advantages of the proposed method, discuss them in conclusion section

5 why deep learning techniques are adequate to deal with condition monitoring tasks under unbalanced data sets?

Author Response

(The authors gave the same response as above.)

Reviewer 3 Report

An improved method of diagnosing rolling bearings based on short-time Fourier transform (STFT), convolutional neural network (CNN), and maximum mean discrepancy (MMD) is presented. As a final step, the fully connected layers were replaced with the adaptive neurofuzzy inference system (ANFIS).

The vibrational signals employed for healthy and defective rolling bearings are from the CWRU database, which is heavily used by many scientists. 

The data acquisition rate of the vibration signal from the drive end (DE) rolling bearing is incorrect. There was mention of 12,000 samples per second, however, it is actually 48,000 samples per second. The results of the current method are unaffected by this.

It is not clear to me why the proposed model was not used as a fault classifier. I'm interested in knowing the predicted results for various faults: inner race, outer race, and ball. The faults from the outer race at 12 o'clock and 3 o'clock cannot be predicted, as I am familiar with the CWRU database. Also, the faults on balls, detected by measurements on the faulty bearing, are hard to predict.

Author Response

(The authors gave the same response as above.)

Round 2

Reviewer 1 Report

The answers to the questions are satisfactory. The article may be published as is.

Reviewer 3 Report

The authors clarified the capabilities of the proposed model and emphasized this aspect in the text of the paper.

I consider that this paper is now clear enough to recommend it for publication.